# Design and Validation of an Instrument for Evaluating Training and Education for Health and Proper Use of Vaccines “VACUNASEDUCA”

**DOI:** 10.3390/ijerph18147321

**Published:** 2021-07-08

**Authors:** Eduardo García-Toledano, Ascensión Palomares-Ruiz, Antonio Cebrián-Martínez, Emilio López-Parra

**Affiliations:** 1Department Public Health, World Rare Disorder Foundation, 28029 Madrid, Spain; toledanoeg@gmail.com; 2Department of Pedagogy, Universidad de Castilla-La Mancha, 02071 Albacete, Spain; antonio.cebrian@uclm.es (A.C.-M.); emilio.lopezparra@uclm.es (E.L.-P.)

**Keywords:** vaccines, questionnaire, education for health, gender, validity, reliability, vaccination schedule, teacher training, COVID-19

## Abstract

The purpose of this article is to demonstrate the design and validation of a questionnaire, to study the importance of vaccines in the health of children. It is based on the analysis of a complete conceptual framework, considering the factors that show the current situation of vaccines, in the context of education for health and coexistence on the international scene. One thousand subjects from 76 countries participated in the study. The validity of the questionnaire was carried out with 15 experts, resulting in a content validity index (IVC) for each high dimension, with the mean IVC being the questionnaire with a Fleiss’ kappa result of 0.96. The process led to a final design consisting of 12 items. The statistic used to identify the degree of consistency in the estimates and calculate the value of the match between the evaluators was the index Fleiss’ kappa, which yielded an overall result of 0.57, which was considered moderate. The results obtained make it possible to conclude that the VACUNASEDUCA questionnaire is valid for understanding the status of vaccines, the importance of teacher training on health prevention, and the influence of student vaccination on the health of peers and teachers, from a gender perspective.

## 1. Introduction

Currently, in the face of the COVID-19 pandemic, social interest has been stoked and research on vaccines has increased. The debate has transcended the context of the experts and flooded the public space, and has been amplified by activism on social media. All this has revealed some contradictions of the experts, confusion of the population, and, in many cases, a decrease in the credibility of vaccines that are affected by a crisis of confidence. It is also being considered whether, in certain cases, they should be mandatory, as well as the administrative and judicial procedure for imposing compulsory vaccination, and the cases in which the courts and tribunals may require it.

It should be specified that this study is based on the fact that health, as indicated in 1946 by the World Health Organization [1], is a state of complete physical, mental and social well-being, and not only as an absence of disease. In this sense, education has a very close relationship with the enjoyment of optimal levels of health. For this reason, internationally renowned organizations, such as the United Nations Educational, Scientific and Cultural Organization (UNESCO), World Health Organization (WHO), United Nations Children’s Fund (UNICEF), the Council of Europe, the Organization for Economic Cooperation and Development (OECD), and the European Commission, indicate that schools need to incorporate health education into their curricula, as a basic tool to develop healthy lifestyle habits, not only to increase the quality of life of schoolchildren, but also to work on building a better, more ecological, tolerant, and supportive world [2]. It should also be stressed that, in the European Union (EU), infectious diseases are a priority issue within public health and, especially, vaccination, which is one of the most effective prevention measures for improving health.

Reasonably, it is essential to analyze the advantages and effectiveness of vaccines, concerns about their safety, and how they are perceived in the educational community. As Matesanz points out [3], having an effective vaccine is not an individual solution, but rather that the maximum number of people in the environment receive it to achieve the desired “herd immunity”, and the virus then stops circulating.

It is clear that vaccines save millions of lives every year, and are one of the safest and most effective public health interventions, providing benefits on disease control and prevention, as well as social and economic benefits. Indeed, since the creation of the Expanded Program on Immunization by the WHO in 1974, the benefits have been expanding, and proof of this is the explicit recognition of the importance of vaccination and its great impact on social good, developing a Global Vaccine Action Plan (GVAP) for 2011–2020 [4], which was approved by 194 countries in the World Health Assembly.

Vaccines are antigenic preparations consisting of microorganisms or by them, which are modified, so that they lose or attenuate their pathogenic capacity, and their purpose is to stimulate the defense mechanisms of the individual against infectious agents. The advances in molecular biology, specifically in the development of recombinant DNA technology and bioinformatics, have been essential tools for the evolution of vaccines. Likewise, the advances in the knowledge of immunity and in biological therapies cannot be forgotten [5]. It is in the twentieth century when the activity gained a greater boom, with the incorporation of biotechnological technical processes. Indeed, from the 1990s to the present, there has been one of the most productive periods in vaccinology, with the achievement of relevant vaccines, such as against avian influenza, retrovirus diarrhea, and hepatitis A and cervical cancer caused by human papillomavirus (HPV).

At present, we live immersed in a situation of continuous virological risk. We are immersed in the pandemic caused by COVID-19, with new infectious challenges and high mortality rates, exacerbating with more virulence diseases, such as malaria, acquired immune deficiency syndrome (AIDS), tuberculosis, etc. This new global framework has brought together and strengthened the role of international bodies. In fact, as it is a public health issue that affects everyone, from the United Nations (UN) and the WHO, strategies and initiatives are being introduced to combat COVID-19, and, at the same time, to be able to continue the work of education, information, monitoring, prevention, and medication worldwide [6].

It must be recognized that, although there is no 100% effective vaccine, nor are all vaccines equally effective, the advantages of vaccines focus on their proven effectiveness, which manifests itself in their behavior in practice and depends, in particular, on the immune capacity of the recipient, the type of vaccine, established schedules, and its availability, tolerance and stability. In the interest of public health, the Spanish Agency of Medicines and Medical Devices (AEMPS) may subject each batch to prior authorization and make the placing on the market conditional on its conformity. Consequently, vaccines are administered in order to stimulate the specific defense mechanisms of the individual against certain infectious agents [7].

The European Union (EU) carries out public health policies in the member states; however, there is no common prevention model. This situation generates inequality in the access to immunization among community citizens, which has been evidenced by the pandemic caused by COVID-19. Each country designs its vaccination strategy, with regard to its epidemiological situation; however, WHO, EU and UNICEF develop prevention and immunization strategies and plans at the global and European level [8].

It is important to emphasize that the WHO designs long-term immunization strategies and carries out specific advocacy work with a global impact, such as World Immunization Week or the European Vaccine Action Plan 2015–2020.

The EU also has a global health strategy involving the AEMPS, the Directorate-General for Health and Food Safety (DG SANTE), and the European Centre for Disease Prevention and Control (ECDC). Thus, the third European action program on health 2014–2020 and the ‘joint actions’ stimulate the collaboration of the member states to solve public health problems at the European level, focusing on the following objectives: to contribute to innovative, effective and sustainable health systems; improve the citizens’ access to better and safer health care; promote health and disease prevention; and protect citizens from cross-border health threats [9].

Our research aims to design and validate a questionnaire to study the importance of vaccines in the health of the child population, to know the influence of vaccines, analyze their development, assess their situation in the twenty-first century, and examine the need for the mandatory nature of certain vaccines.

Reasonably, it was proven that there is no questionnaire that is adequate to what is intended to be measured in our research, so it was essential to create a new questionnaire, VACUNASEDUCA [2], following the steps we recommend for the process of the creation and validation of questionnaires by various authors [10,11,12,13,14].

## 2. Materials and Methods

The analytical descriptive design adopted by the questionnaire validation process has been developed in several phases in which quantitative strategies and the application of different statistical techniques have predominated through the IBM SPSS26 program, for the calculation of descriptive statistics, reliability indices through Cronbach’s alpha model and the exploratory study of the factorial structure through the factor analysis of principal components. The results of these analyses have been the subject of quantitative assessments by 15 experts (Table 1) using an ad hoc questionnaire. To evaluate the content validity, the Lawshe content index was calculated for each item using a Likert scale with a score of 1 to 3 and to evaluate the degree of agreement among experts, the Fleiss’ kappa concordance coefficient was calculated using a Likert scale with a score between 1 to 5 with respect to 3 criteria (clarity, relevance, and significance).

### 2.1. Participants

The sample of this research, given the subject matter studied, has been worldwide. It has been considered as the subset of individuals who agree to undergo an investigation, where it is taken as a reference to meet several criteria of representativeness of the total population. The selection has been in the interest of the researchers. Indeed, the sample consists of 1000 participants from the following 76 countries: Albania, Germany, Andorra, Angola, Saudi Arabia, Algeria, Argentina, Australia, Austria, Bangladesh, Belgium, Bolivia, Bosnia-Herzegovina, Spain, Botswana, Brazil, Bulgaria, Cape Verde, Cameroon, Canada, Chile, China, Cyprus, Colombia, Ivory Coast, Cuba, Denmark, Ecuador, United Arab Emirates, Spain, United States, Estonia, Philippines, Finlandia, France, Gabon, Gambia, Georgia, Greece, Guatemala, Equatorial Guinea, Haiti, Honduras, India, Ireland, Iceland, Israel, Italy, Jamaica, Japan, Jordan, Latvia, Lebanon, Liberia, Luxembourg, Morocco, Mauritius, Mauritania, Mexico, Montenegro, Mozambique, Paraguay, Peru, Poland, Portugal, United Kingdom, Russia, South Africa, Sweden, Switzerland, Thailand, Tanzania, Turkey, Ukraine, Uganda, Venezuela, Zimbabwe.

Regarding the characteristics of the sample, in relation to gender, it is made up of 694 women (69.4%) and 306 men (30.6%). The age of the sample members is mostly between 18 and 44 years (Figure 1).

As for the work performance of the sample members, 55.4% (*n* = 554) belong to the health sector, 32.9% (*n* = 329) to the educational field and 11.7% (*n* = 117) to the economy sector.

The socioeconomic level of the sample members has been established based on the country of origin through the human development index (HDI) (Figure 2).

### 2.2. Instrument

As indicated, the conceptual framework of the VACUNASEDUCA questionnaire focuses on vaccines, being configured in the main variable of research, its structure is reflected in Table 2. Each item was evaluated using a Likert scale with a rating between 1 and 3: No (1), Maybe (2), Yes (3).

### 2.3. Procedure

The research has been developed in the following phases: (a) delimitation of the construct to be measured and analysis of the state of the matter; (b) pilot design of the questionnaire, taking into account the objective, population, sample and format; (c) expert judgement who will assess the validity of the content of the questionnaire and make the relevant proposals for improvement and its incorporation; (d) statistical analysis of the data; (e) final proposal for the questionnaire. The study was carried out in 2019 and 2020, guaranteeing at all times the confidentiality and anonymity of the data obtained (Appendix A).

## 3. Results

The statistical results obtained in the calculation of the content validity, the construct validity, and the reliability of the final instrument are shown.

### 3.1. Validity of Content

In order to determine the validity of content, the Lawshe’s content index [15] has been used. Calculating the content validity ratio (CVR) for each time specifies which items of the instrument are appropriate and, therefore, must remain in the final version of the questionnaire. A score must be determined for each item according to the following three possibilities: that the element is fundamental to evaluate the construct; that it is useful, but dispensable; or that it is considered unnecessary. Subsequently, the content validity index (CVI) is calculated for the instrument as a whole, which is no more than a mean of the content validity of all the items. Lawshe himself suggests that CVI = 0.51 when 14 experts have been used.

This verifies the extent to which the questionnaire actually measures what you want to measure, and thus assesses the consistency of the questionnaire, debugging its initial structure.

The affirmative answers given by the participating experts, as reflected in Table 3, were taken into consideration for the calculation of the CVR.

As can be seen, high content validity ratios were found in all the items, with no value below 0.51 appearing, so there was no need to remove any item from the initial questionnaire. It should be noted that 10 items, 83.3% of the total of the 12 items, achieved a perfect content validity ratio; a value of one and only two items, 16.7% of the total, obtained a ratio of 0.73, which was indicative of a high content validity [16]. The indices for each dimension were high, with the mean rate being 0.96.

The value of the agreement between the evaluators was also calculated, in order to know the degree of consistency in the estimates made. The kappa coefficient, originally proposed by Cohen [17], is used to assess the degree of agreement between two evaluators; although in this research the generalized version, known as Fleiss’ kappa index, was applied, because it is being used for multiple experts (more than three), based on the concept of notion of agreement among several observers [18,19]. Each item of the initial questionnaire was evaluated by the experts on a Likert scale, with a score between one and five, as follows: nothing (1), little (2), regular (3), enough (4), and much (5), in respect to three criteria (clarity, relevance, and significance). For the interpretation of Fleiss’ kappa coefficient, the scale established by Landis and Koch [20] was taken into account, which qualitatively expresses the strength of agreement between the evaluators, as follows: 0.00–0.09, poor; 0.10–0.20, slight; 0.21–0.40, fair; 0.41–0.60, moderate; 0.61–0.80, substantial; and 0.81–1.0, almost perfect.

Table 4 shows that, together, the instrument obtained a 0.57 moderate match in the expert trial, highlighting the Fleiss’ kappa coefficient with respect to the relevance criterion, with a considerable 0.61 match.

### 3.2. Construct Validity

#### 3.2.1. Exploratory Factor Analysis to Study Construct Validity

The preliminary tests of the factor analysis allow us to verify the adequacy of the factorial technique, these being the following: the Kaiser–Meyer–Olkin test (KMO) and the Bartlett Sphericity test [21,22,23,24]. In Table 5, it can be verified that the KMO index obtained is 0.784, revealing the sample adequacy for the performance of the factor analysis. The results of Bartlett’s Sphericity test yield a significance level of 0.000, which implies the ability to reject the null hypothesis and initiate factor analysis.

After validation by expert judgment, the instrument was configured by 12 items that were included in the factor analysis.

The results of the extraction method, by analyzing the principal components, are reflected in Table 5.

The analysis in Table 6 allows us to continue the factor analysis with four main factors, with a percentage of total variance explained of 67.399%, which is a reasonably acceptable variability.

Most of the factors studied in the social sciences are interrelated, so externally imposing a criterion of non-correlation between factors, such as orthogonal rotations, may be artificial and not respond adequately to reality. Consequently, it has been chosen to use oblique rotations that increase the realism of the factorial solution, so the Oblimin factorial rotation method is used, with Kaiser normalization [25], obtaining the results shown in Table 6.

The matrix of components did not come out totally “clean”, with items three and eight correlating in two factors.

The factors, F1, F2, F3 and F4, coincide with the dimensions and their items described preliminarily before the factorial analysis procedure: D1, awareness and regulation; D2, education and teachers; D3, regulation and mandatory; and D4, consequences and risks. Although item three correlates with factors F1 and F2, and item eight correlates with factors F3 and F4, in general, a factorial structure that is very similar to that initially designed can be observed.

#### 3.2.2. Descriptive Statistics for Each of the Items and Dimensions

The descriptive statistics for each of the items are shown in Table 7. It can be observed that the items with the highest averages are from 1 to 6, and the items with the lowest averages are from 7 to 12.

For each of the dimensions, its descriptive statistics are collected in Table 8. There is a higher average score for dimensions D1, awareness and regulation, and D2, education and teachers, with the score being lower for dimensions D3, regulation and mandatory, and D4, consequences and risks. The mean of the four dimensions was 2.05 (DT = 0.22).

#### 3.2.3. Calculation of the Reliability of the Final Instrument

With the calculation of reliability, we tried to analyze the stability of the results for future administrations of the questionnaire, using Cronbach’s Alpha coefficient (α).

The reliability analysis for the instrument reached an index of 0.64 for all the items on the questionnaire. Table 9 specifies Cronbach’s Alpha for each of the dimensions of the questionnaire.

The mean value for the four dimensions was 0.64, which is close to the limit of 0.70 that Kerlinger et al. [26] set for acceptable consistency. In short, good consistency of the whole instrument is evident.

## 4. Conclusions

In health education, it is evident that the use of reliable and validated questionnaires is a very common methodological instrument that requires adequate elaboration, as it is a fundamental part of the research design and conditions the results obtained [14,21,27]. Reasonably, this article has shown the process followed for the validation of the questionnaire VACUNASEDUCA, which was developed with the purpose of deepening the importance of vaccines.

It should be noted that the work began before the pandemic caused by COVID-19, being topical and highly applicable to the new challenges in vaccination in the educational context, especially in the first levels of schooling. There is no doubt that, today, one of the most worrying challenges is global health security. In addition, it is becoming clear with COVID-19 that disease outbreaks can develop faster and travel farther than ever before. Emerging threats are not limited to a single place or country, so investing in health security must be a global priority.

It should also be noted that we live in a healthier, safer, and more prosperous world than 20 years ago; however, the progress made is fragile. Indeed, more than one and a half million people die each year from vaccine-preventable diseases, and fifteen million children are still missing out on the benefits of vaccination in the least developed countries. Along these lines, the Vaccine Alliance (GAVI) has planned that, in the strategic period 2021–2025, the launch of the most complete package of vaccines in the history of the Alliance will take place, expanding the portfolio of vaccines. Logically, the introduction of new vaccines, including boosters against tetanus, diphtheria, and pertussis (Tdap); the birth dose of hepatitis B; multivalent meningococcal conjugate vaccines; routine oral cholera vaccine; and vaccines against respiratory syncytial virus (RSV) and rabies will help protect people throughout their lives, not just during childhood [2].

There is no doubt that global health security and the fight against COVID-19 depend on the early implementation of health systems around the world. Likewise, actions, treatments, and efforts are needed to combat this pandemic, where health education from the earliest age is of vital importance.

It must be recognized that, in order to motivate the realization of work on the subject at hand, and to be able to develop solid and effective research, it is essential to have valid tools that make it possible for the results obtained in the research to be useful to intervene in the effective improvement of the situation studied. There is no doubt that knowing what their political, social and human factors are allows us to investigate in a little-known field, promoting the study of the concerns and opinions of the human factors involved in a more direct way, such as teachers at university and non-university centers, scientists, parents and mothers, teachers in training, students, political representatives, and managers. Consequently, the researchers prioritized designing a questionnaire that was very easy to complete, in order to optimize the response of potential participants.

The VACUNASEDUCA questionnaire can be a very valuable tool in today’s society, and, as has been shown, it is easy to apply and requires few prerequisites. It is justified that it is very valid to know, in a given context, the awareness of society regarding vaccines, the assessment of their mandatory nature, the regulation by the administrations, the initial training and awareness of teachers, gender influence, information and involvement of parents, need for a vaccination register/card, and the consequences and risks of non-vaccination. It is a questionnaire of great interest not only in the educational and health fields, but it also affects the whole of society, because vaccines are an important measure, due to their high effectiveness and tolerance profile, making it possible to save millions of lives and eradicate some diseases. There is no doubt that vaccines are an investment not only because they improve the quality of life, but they also constitute an economic and social value of great importance.

It offers an original and innovative proposal that, based on the training offered to teachers on health prevention, plans to design a training plan for prevention and education for health, and reflect on the influence of the vaccination of students on the occupational health of teachers. In addition, from a gender perspective, it is possible to analyze how the effects of the inappropriate application of vaccines could affect male teachers, differently from female teachers. 

It should be noted that one of the strengths of the VACUNASEDUCA questionnaire has been the very large sample used, which covers 76 countries. It is also important to note that the process of validating a questionnaire has limitations linked to the choice of the number of experts, since their number varies substantially according to the authors consulted, without establishing a specific consensus [28]. Thus, there are authors who indicate 3 as a valid number of experts, while others establish a range between 14 and 25 experts [29]. As indicated, 15 experts participated in the validation of the VACUNASEDUCA questionnaire, which is a number within the ranges established by different authors. Also, a significant strength of the study is the methodological rigor practiced both in the content validation process and in the internal validation process. Therefore, it can be concluded that a questionnaire is available with sufficient internal validity and reliability, and is easy to apply in different contexts and areas of society at the international level.

## Figures and Tables

**Figure 1 ijerph-18-07321-f001:**
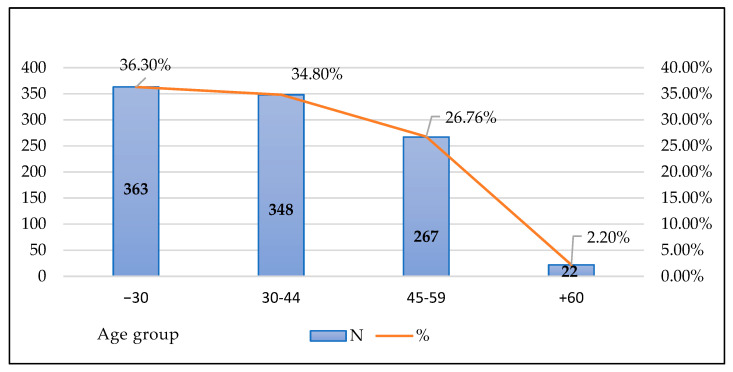
Characteristics of the sample in relation to age.

**Figure 2 ijerph-18-07321-f002:**
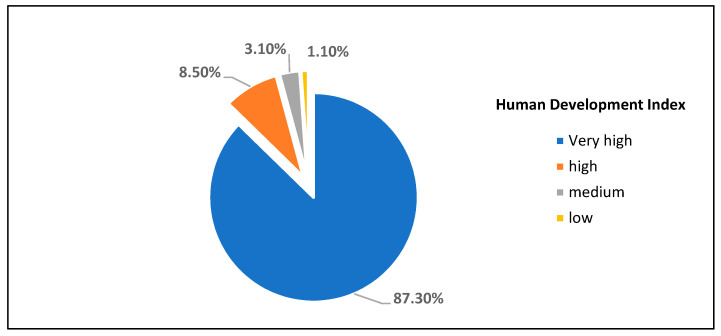
Number of respondents by human development index.

**Table 1 ijerph-18-07321-t001:** Profession and gender of experts.

N.º	Profession	Gender	Years of Experience	Sector
1	Director of the immunology area of University Clinical Hospital	Man	+20	Health
2	University Professor in Microbiology	Man	+20	Health
3	University Professor	Woman	+20	Economy
4	Specialist in Pulmonology and University professor	Man	+20	Health
5	Teacher of Secondary Education and University	Woman	10–20	Education
6	University Professor	Man	+20	Economy
7	Investigator	Woman	−10	Economy
8	Full Professor University	Woman	+20	Education
9	Nursery and Primary Teacher	Man	−10	Education
10	Primary Teacher	Woman	10–20	Education
11	Children’s Teacher	Woman	10–20	Education
12	Master of Therapeutic Pedagogy	Man	10–20	Education
13	Professor at University	Man	+20	Education
14	Primary Education Counselor	Woman	−10	Education
15	University Professor	Woman	+20	Education

Source: own elaboration.

**Table 2 ijerph-18-07321-t002:** Dimensions and items of the questionnaire.

Dimensions	Items
(D1) Awareness and regulation	P01. Do you value that society is aware of the importance of vaccination?
	P02. Do you believe that administrations have regulated actions for vaccine compliance?
(D2) Education and teachers	P03. Do you consider that, in your country, the training of compulsory education teachers (children and primary) provides adequate training on vaccines?
	P04. Do you think teachers are aware of the proper use of vaccines?
	P05. Do you think parents know the consequences that coexistence with other non-vaccinated peers could have on their children?
	P06. Do you believe that teachers at mandatory levels should receive initial training on health education and specifically on the vaccination process?
(D3) Regulation and enforceability	P07. Do you think it is necessary for teachers to require students to have a scheduled vaccination card?
	P08. Do you know if there is adequate regulations to support teachers in the demand to comply with childhood vaccination?
(D4) Consequences and risks	P09. Do you think teachers know the consequences that coexistence with non-vaccinated children could have on students and their person?
	P10. Do administrations have measures to ensure the health of pregnant teachers?
	P11. Do administrations have measures to ensure the health of teachers with minor children or family members with at-risk diseases?
	P12. Do parents of vaccinated students know the risk of their children when living with other non-vaccinated peers?

Source: own elaboration.

**Table 3 ijerph-18-07321-t003:** Content validity reason (CVR) of items.

Dimension	Item	No	Maybe	Yes	CVR
Awareness and regulation	1	0	2	13	0.73
	2	0	0	15	1
Education and teachers	3	0	0	15	1
	4	0	0	15	1
	5	0	2	13	0.73
	6	0	0	15	1
Regulation and enforceability	7	0	0	15	1
	8	0	0	15	1
Consequences and risks	9	0	0	15	1
	10	0	0	15	1
	11	0	0	15	1
	12	0	0	15	1

Source: own elaboration.

**Table 4 ijerph-18-07321-t004:** Fleiss’ kappa coefficient and statistical significance of the characteristics of the original instrument.

Characteristics	Fleiss’ Kappa Coefficient	Z	Sig.
Clarity	0.53	31.71	0.000
Pertinence	0.58	32.54	0.000
Relevance	0.61	34.29	0.000
Total	0.57		

Source: own elaboration.

**Table 5 ijerph-18-07321-t005:** Total variance explained from the main components.

Total Variance Explained							
Component	Initial Eigenvalues			Extraction Sums of Squared Loadings			Rotation Sums of Squared Loadings
	Total	% of Variance	Cumulative %	Total	% of Variance	Cumulative %	Total
1	3.864	32.204	32.204	3.864	32.204	32.204	3.678
2	1.939	16.156	48.359	1.939	16.156	48.359	1.858
3	1.209	10.074	58.433	1.209	10.074	58.433	1.614
4	1.076	8.966	67.399	1.076	8.966	67.399	1.465
5	0.903	7.521	74.920				
6	0.727	6.060	80.980				
7	0.607	5.058	86.037				
8	0.576	4.802	90.839				
9	0.507	4.221	95.060				
10	0.295	2.460	97.520				
11	0.197	1.639	99.160				
12	0.101	0.840	100.000				

Note. Extraction method: principal component analysis. When components are correlated, sums of squared loadings cannot be added to obtain a total variance.

**Table 6 ijerph-18-07321-t006:** Matrix of rotated components for four factors.

Pattern Matrix A				
Items	Component			
	1	2	3	4
P01			0.776	
P02			0.811	
P03		0.577	0.537	
P04		0.741		
P05		0.732		
P06		0.524		
P07				0.794
P08	0.593			0.649
P09	0.827			
P10	0.934			
P11	0.932			
P12	0.882			

Note. Extraction method: principal component analysis. Rotation method: Oblimin with Kaiser normalization. Rotation converged in 10 iterations.

**Table 7 ijerph-18-07321-t007:** Descriptive statistics for each item.

Descriptive Statistics									
Items	N	Min.	Max.	Mean	Std. Deviation	Skewness		Kurtosis	
Items	Statistic	Statistic	Statistic	Statistic	Statistic	Statistic	Std. Error	Statistic	Std. Error
P_01	1000	1	3	2.77	0.477	−2.007	0.077	3.276	0.155
P_02	1000	1	3	2.84	0.403	−2.478	0.077	5.632	0.155
P_03	1000	1	3	2.8	0.477	−2.395	0.077	5.017	0.155
P_04	1000	1	3	2.79	0.467	−2.219	0.077	4.243	0.155
P_05	1000	1	3	2.79	0.485	−2.301	0.077	4.543	0.155
P_06	1000	1	3	2.86	0.44	−3.153	0.077	9.245	0.155
P_07	1000	1	3	1.59	0.803	0.888	0.077	−0.874	0.155
P_08	1000	1	3	1.36	0.59	1.428	0.077	0.988	0.155
P_09	1000	1	3	1.24	0.505	1.976	0.077	3.082	0.155
P_10	1000	1	3	1.2	0.451	2.265	0.077	4.502	0.155
P_11	1000	1	3	1.18	0.424	2.341	0.077	4.904	0.155
P_12	1000	1	3	1.18	0.427	2.293	0.077	4.652	0.155

Source: own elaboration.

**Table 8 ijerph-18-07321-t008:** Descriptive statistics for questionnaire dimensions.

Descriptive Statistics									
Items	N	Min.	Max.	Mean	Std. Deviation	Skewness		Kurtosis	
	Statistic	Statistic	Statistic	Statistic	Statistic	Statistic	Std. Error	Statistic	Std. Error
D1 Awareness and regulation	1000	1.00	3.00	2.8070	0.36247	−1.697	0.077	1.729	0.155
D2 Education and teachers	1000	1.50	3.00	2.8105	0.30689	−1.845	0.077	3.161	0.155
D3 Regulation and enforceability	1000	1.00	3.00	1.4715	0.58973	0.939	0.077	−0.255	0.155
D4 Consequences and risks	1000	1.00	3.00	1.1993	0.40608	2.136	0.077	4.050	0.155
TOTAL	1000	1.33	3.00	2.0497	0.21669	1.068	0.077	2.445	0.155

Source: own elaboration.

**Table 9 ijerph-18-07321-t009:** Cronbach Alpha by dimension and its items.

Dimension	Items	A
D1 Awareness and regulation	1, 2	0.52
D2 Education and teachers	3, 4, 5, 6	0.56
D3 Regulation and enforceability	7, 8	0.57
D4 Consequences and risks	9, 19, 11, 12	0.92
Mean		0.64

Source: own elaboration.

## Data Availability

The data presented in this study are available in Appendix A.

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
