# Peer review of "Design and Validation of an Instrument for Evaluating Training and Education for Health and Proper Use of Vaccines “VACUNASEDUCA”"

_ijerph, 2021, doi:10.3390/ijerph18147321_

Round 1
Reviewer 1 Report
The bibliographic sources consulted must follow a citation orderAuthor Response
First of all, we would like to thank the reviewers for their contributions which have undoubtedly contributed significantly to the quality of the article.
REVIEWER #1
The bibliographic sources consulted must follow a citation order.
ANSWER: The bibliographical references have been reviewed and sorted according to their citation in the text.
Reviewer 2 Report
The research contributes to healthcare studies, particularly on the importance of vaccination on children´s health.
The research covers an important number of subjects (1000) from different countries. This contributes to a strong methodology and a wider comparative specter, furthermore it obtains good validity indexes and instrument reliability.
The results corroborate the utility of the instrument, and the conclusion demonstrates the possibility of continuing applying the instrument in other contexts to obtain valid and trustworthy results.
Finally, it is considered necessary to adjust the following references.
2,
3,
4,
5,
16,
26.
Author Response
First of all, we would like to thank the reviewers for their contributions which have undoubtedly contributed significantly to the quality of the article.
REVIEWER #2
The research contributes to healthcare studies, particularly on the importance of vaccination on children´s health.
The research covers an important number of subjects (1000) from different countries. This contributes to a strong methodology and a wider comparative specter, furthermore it obtains good validity indexes and instrument reliability.
The results corroborate the utility of the instrument, and the conclusion demonstrates the possibility of continuing applying the instrument in other contexts to obtain valid and trustworthy results.
Finally, it is considered necessary to adjust the following references 2, 3, 4, 5, 16, 26.
ANSWER: All the references indicated have been reviewed and adjusted appropriately:
Reviewer 3 Report
the article Design and Validation of an Instrument for Evaluating Training and Education for Health and Proper Use of Vaccines“VACUNASEDUCA”
is considered to be of great relevance and interest to the subject matter of the journal.
It,s presented as a research text of interest to the readers and editors of the journal.
I consider that a strong revision of the English wording should be made, as well as an improvement of the conclusions (less general) and more concrete about the data obtained.
In short, the text should be generally revised both in the wording and in the content related to the theoretical background, as well as in the conclusions and discussion.
Author Response
First of all, we would like to thank the reviewers for their contributions which have undoubtedly contributed significantly to the quality of the article.
REVIEWER #3
It is considered to be of great relevance and interest to the subject matter of the journal.
It Is presented as a research text of interest to the readers and editors of the journal.
I consider that a strong revision of the English wording should be made, as well as an improvement of the conclusions (less general) and more concrete about the data obtained.
In short, the text should be generally revised both in the wording and in the content related to the theoretical background, as well as in the conclusions and discussion.
ANSWER: All the indications by the reviewer have been followed.
Reviewer 4 Report
Dear Authors,
I attached my review. The topic is relevant and there are important practical implications, but I find manuscript in its current form unsuitable for publishing. Please find my comments and suggestions in the attached file.

Author Response
First of all, we would like to thank the reviewers for their contributions which have undoubtedly contributed significantly to the quality of the article.
REVISOR #4
- Firstly, there are too many missing information on sample that need to be in the paper. Apart from a total number of participants (N = 1000) and countries they are coming from, we know nothing about the sample age, gender, education, socioeconomic background etc. It is not clear who the participants were. This needs to be improved before considering paper for publication. Furthermore, we do not have information on subsample of experts (only gender); they were experts in what area? What is their expertise?
ANSWER: Modifications have been introduced detailing characteristics of the sample related to age, gender, and socioeconomic status.
Regarding the characteristics of the sample, in relation to gender, it is made up of 694 women (69.4%) and 306 men (30.6%). The age of the sample members is mostly between 18 and 44 years (Figure 1).
Figure 1. Characteristics of the sample in relation to age.
As for the work performance of the sample members, 55.4% (n=554) belong to the health sector, 32.9% (n=329) to the educational sector and 11.7% (n=117) to the economy sector.
The socioeconomic level of the sample members has been established based on the country of origin through the Human Development Index (HDI) (Figure 2).
Figure 2. Number of respondents by Human Development Index.
- In addition, we have no information on the subsample of experts (gender only); were they experts in what area? What is your experience?
ANSWER: Additional information on the expert sub-sample (Profession, Gender, Sector and Experience) has been added.
Table1. Profession and gender of experts
|
N.º |
Profession |
Gender |
Years of Experience |
sector |
|
1 |
Director of the immunology area of University Clinical Hospital |
Man |
+20 |
Health |
|
2 |
University Professor in Microbiology |
Man |
+20 |
Health |
|
3 |
University Professor |
Woman |
+20 |
Economy |
|
4 |
Specialist in Pulmonology and University professor |
Man |
+20 |
Health |
|
5 |
Teacher of Secondary Education and University |
Woman |
10-20 |
Education |
|
6 |
University Professor |
Man |
+20 |
Economy |
|
7 |
Investigator |
Woman |
-10 |
Economy |
|
8 |
Full Professor University |
Woman |
+20 |
Education |
|
9 |
Nursery and Primary Teacher |
Man |
-10 |
Education |
|
10 |
Primary Teacher |
Woman |
10-20 |
Education |
|
11 |
Children's Teacher |
Woman |
10-20 |
Education |
|
12 |
Master of Therapeutic Pedagogy |
Man |
10-20 |
Education |
|
13 |
Professor at University |
Man |
+20 |
Education |
|
14 |
Primary Education Counselor |
Woman |
-10 |
Education |
|
15 |
University Professor |
Woman |
+20 |
Education |
Source: own elaboration.
- In the procedure part, there is no information on how the starting items were created.
ANSWER: The research team has experience in the design and validation of other questionnaires, after the corresponding previous work, properly assessed the importance of having a questionnaire that was easy to complete with clear and concise language for people from various areas of society and countries of the world and could be carried out in a short time. Therefore, after carefully studying the various options, the questionnaire was configured by 12 items grouped in 4 dimensions, which were submitted for the consideration of the 15 experts.
- What the instrument is intended to measure: this information is missing or unclear.
ANSWER: The article clearly specifies. As a clarification to the reviewer, he is informed that the questionnaire VACUNASEDUCA is very useful to know how the awareness of society about vaccines is perceived and the focus of the interviewees on the regulation of administrations for compliance. It also measures the perception of the importance of education and the role, training, and involvement of teachers in vaccination. In addition, it studies a topic of the utmost topicality and interest throughout the world, such as the importance of the mandatory nature of certain vaccines. No less important is the information that the questionnaire can provide on the knowledge of the consequences and risks that administrations, teachers and parents have on the proper use of vaccines. In summary, VACUNASEDUCA is a questionnaire that provides valuable information very useful in today's society that can guide to improve health education for all people.
- Regarding the objective of the study, the authors wrote that "the objective was to study the importance of vaccines in the health of children, analyze the development of the application of vaccines, know and evaluate their situation in the twenty-first century" .... this goes far beyond the objectives of the study; nor is it clear what the authors mean by this formulation; in short, this was not the aim of the study and needs to be improved.
ANSWER: The objective of the research has been clarified:
Our research aims to design and validate a questionnaire to study the importance of vaccines in the health of the child population to know the influence of vaccines, analyze their development, assess their situation in the twenty-first century and examine the need for the mandatory nature of certain vaccines.
- The author stated that the design of the research was quantitative and quasi-experimental; I am not sure this is the correct description.
ANSWER: The description of the research design has been modified.
The analytical descriptive design adopted by the questionnaire validation process has been developed in several phases in which quantitative strategies and the application of different statistical techniques have predominated through the IBM SPSS26 program, for the calculation of descriptive statistics, reliability indices through Cronbach's Alpha model and the exploratory study of the factorial structure through the factor analysis of principal components. The results of these analyses have been the subject of quantitative assessments by 15 experts (Table 1) using an ad hoc questionnaire. To evaluate the content validity, the Lawshe content index was calculated for each item using a Likert scale with a score of 1 to 3 and to evaluate the degree of agreement among experts, the Fleiss Kappa concordance coefficient was calculated using a Likert scale with a score between 1 to 5 with respect to 3 criteria (clarity, relevance, and significance).
- It is not clear why the authors started with only 12 items, and even less why they divided them a priori into four categories. It is usual to start with more items and, after all the procedures, take them to the number that is optimal to cover the construct, and that is not too long for the administration. Why were there only two items for two dimensions/factors: awareness and regulation; regulation and applicability? The experts subsequently evaluated all the items as relevant and clear, so the authors kept them all and calculated the reliability in two items per factor, which is also a major drawback. In addition, the reliability of three of the four factors was too low. However, the authors conclude that the instrument is reliable, but it is not clear on what basis it is based.
ANSWER: Table 9 shows the indices obtained for each dimension highlighting the index of .92 for dimension D4 Consequences and risks. The mean value for the 4 dimensions has been .64 being close to the limit .70 that Kerlinger et al. [26] set for acceptable consistency.
The research team has experience in the design and validation of other questionnaires, after the corresponding previous work, properly assessed the importance of having a questionnaire that was easy to complete with clear and concise language for people from different areas of society and countries of the world and could be carried out in a short time. Therefore, after carefully studying the various options, the questionnaire was configured by 12 items grouped in 4 dimensions, which were submitted for the consideration of the 15 experts.
- The document should reduce the introductory part and strengthen the discussion, which is very brief in terms of the interpretation of the results.
ANSWER: Changes have been made to the introduction and the discussion has been broadened.
- It is also unnecessary to present the same data more than once, so in that sense some parts should be omitted, for example, table 3 (information already written in the text), table 5 (already written in the text), the introductory part of subchapter 3.2.1...; the question is whether a screen plot is necessary when the eigenvalues are written in table 6; table 8 is unnecessary because everything is visible in table 7, etc.
ANSWER: The considerations indicated by the reviewer have been taken into account.
Round 2
Reviewer 3 Report
DEAR AUTHORS, ALTHOUGH THE PUBLICATION HAS BEEN CONSIDERED, IT IS APPRECIATED THAT THE AUTHORS DO NOT HAVE OR ARE NOT ALIGNED AS RESEARCHERS WITH THE SUBJECTS OBJECT OF THE STUDY, NOR WITH THE MODEL / METHODOLOGICAL FRAMEWORK. THIS MAKES IT LESS ALIGNED WITH THE CALL FOR PAPERS THEME AND HAVE LESS IMPACT OR TRANSFER OF RESULTS. THE AUTHORS AND THE ORDER THAT REFLECTS SHOULD SUPPORT THE INVESTIGATION BY THEIR ACADEMIC AND RESEARCH CAREER.Author Response
REVISOR #3
DEAR AUTHORS, ALTHOUGH THE PUBLICATION HAS BEEN CONSIDERED, IT IS APPRECIATED THAT THE AUTHORS DO NOT HAVE OR ARE NOT ALIGNED AS RESEARCHERS WITH THE SUBJECTS OBJECT OF THE STUDY, NOR WITH THE MODEL / METHODOLOGICAL FRAMEWORK. THIS MAKES IT LESS ALIGNED WITH THE CALL FOR PAPERS THEME AND HAVE LESS IMPACT OR TRANSFER OF RESULTS. THE AUTHORS AND THE ORDER THAT REFLECTS SHOULD SUPPORT THE INVESTIGATION BY THEIR ACADEMIC AND RESEARCH CAREER.
ANSWER The reviewer's instructions have been heeded. In addition, the reviewer's ratings are appreciated and will be taken into consideration for future work. Just for indication, we inform you that the first author in Doctor of Medicine and Education (linked to the WHO), and the other three are working and researching in education, since the questionnaire unites research in education and health (VACUNASEDUCA). The team has produced numerous publications of the research carried out.
